# Clinically Meaningful Change in 6 Minute Walking Test and the Incremental Shuttle Walking Test following Coronary Artery Bypass Graft Surgery

**DOI:** 10.3390/ijerph192114270

**Published:** 2022-11-01

**Authors:** Suman Sheraz, Humera Ayub, Francesco V. Ferraro, Aisha Razzaq, Arshad Nawaz Malik

**Affiliations:** 1Faculty of Rehabilitation and Allied Health Sciences, Riphah International University, Islamabad 46000, Pakistan; 2School of Human Sciences, College of Science and Engineering, University of Derby, Derby DE22 3AW, UK

**Keywords:** cardiac rehabilitation, exercise tolerance, functional mobility, innovation in rehabilitation, physical activity

## Abstract

The 6-min walk test (6MWT) and incremental shuttle walk test (ISWT) are widely used measures of exercise tolerance, which depict favorable performance characteristics in a variety of cardiac and pulmonary conditions. Both tests are valid and reliable method of assessing functional ability in cardiac rehabilitation population. Several studies have calculated the minimal clinically important difference (MCID) of these exercise tests in different populations. The current study aims to estimate MCID of 6MWT and ISWT in patients after Coronary artery bypass graft (CABG) surgery. In this descriptive observational study, nonprobability purposive sampling technique was used to assess 89 post CABG patients. The participants performed the 6MWT and ISWT along with vital monitoring on third, fifth and seventh post operative days. The data was with calculation of 6MWT and ISWT MCID through distribution and anchor-based methods. Results showed significant improvement (*p* < 0.001) in 6MWT as well as in ISWT after seven days of in-patient cardiac rehabilitation. The minimal detectable difference of 6MWT determined by the distribution-based method was 36.11 whereas MCID calculated by Anchor based method was 195 m. The minimal detectable difference of ISWT determined by the distribution-based method was 9.94 whereas MCID calculated by Anchor based method was 42.5 m. In conclusion our results will assist the future researchers and clinicians to interpret clinical trials as well as to observe the clinical course of post operative cardiac patients.

## 1. Introduction

Coronary Artery Disease (CAD) is a leading cause of mortality and morbidity worldwide [1]. Cardiac Rehabilitation (CR) is a multidisciplinary comprehensive program comprising of exercise training, counselling, and patient education; designed for the patients with CAD as well as after the revascularization procedures like coronary artery bypass graft surgery (CABG), both of which are associated with a decline in functional status, exercise tolerance, and aerobic capacity [2].Exercise training is a major component of CR programs as it improves hemodynamic responses [3], functional capacities [2] as well as quality of life after CABG [4,5]. For the assessment of cardiopulmonary endurance (i.e., exercise tolerance), multiple laboratories and objective assessments (such as walking tests) are used [6].

Laboratory based tests, such as peak oxygen consumption, are gold standard but they are costly, impractical and difficult to perform in post operative cardiac surgery patients [7]. Therefore, objective assessments (e.g., walk tests) including maximal and sub maximal tests are employed for the assessment of exercise tolerance of these patients [8]. These walk tests are easy to administer, low at cost, and incorporate simple functional activity, to which the patients are already familiar with. Additionally, the technical expertise required to perform these tests is less complex compared to the laboratory-based tests [9].

Therefore, 6 Minute Walk Test (6MWT) which is sub-maximal and Incremental Shuttle Walk Test (ISWT)—a maximal test is widely used for assessment of functional capacity and exercise tolerance of post-CABG patients [10,11,12]. Clinical trials reporting statistically significant changes in these outcome measures clarify that the results of trial or effectiveness of treatment is not by chance. However, they fail to clarify that this statistically significant improvement is clinically evident and meaningful to the patient [9,13]. The concept used to quantitatively determine the amount of change in a clinical outcome which is clinically meaningful to the patient is Minimum Clinically Important Difference (MCID) [13], defined as the smallest difference in an outcome measure which is perceived as important and beneficial to the patient [14]. Changes in an outcome measure greater than the threshold MCID indicates that the improvements measured are beneficial, and meaningful for that patient.

Multiple approaches are used in literature for calculating the MCID of outcome measures [15]. The two main approaches are distribution-based and anchor-based [16]. Distribution based approach uses the statistical characteristics of the obtained sample to estimate MCID [17]. The methods in this approach include calculation of standard error of measurement (SEM), effect size (ES) and minimal detectable difference (MDD) (18). Whilst the anchor-based methods compare the changes in patient’s score with another external measure of change labelled as anchor which can be either subjective or objective [18,19]. The MCID threshold used by Busch et al. for CABG patients attending CR was 54 m [20,21]. Whilst there is no reported value of MCID of 6MWT after CABG till date, the calculated MCID of ISWT after CR program attended by post myocardial infarction, post percutaneous coronary intervention as well as post CABG patients is 70 m [9].

Since, simultaneous use of both distribution and anchor-based approaches is recommended to determine effect of the method on final value [19], the current study aims to calculate the MCID for 6MWT and ISWT after in-patient cardiac rehabilitation utilizing both distribution-based and anchor-based approaches with the intention of developing better way of understanding and improvement in health care goals for post-CABG patients.

## 2. Materials and Methods

### 2.1. Study Design

An observational study was conducted in Rawal General Hospital and Armed Force Institute of Cardiology (Rawalpindi, Pakistan) with the aim of calculating MCID for 6MWT and ISWT after cardiac rehabilitation using both distribution and anchor-based approaches. The calculated sample size for study is 83 calculated with power analysis at the power of 80% and a significance level of 5% [22]. To account for dropouts, as much as available participants were recruited. Thus, the achieved sample size was 92 participants, out of which 3 dropped out and 89 participants completed the protocol The protocol was approved by the Riphah Ethical Committee (REC/00888) and was conducted in accordance with the declaration of Helsinki [23]. Participants were selected via non-probability purposive sampling technique and an informed consent was obtained in written form before initiating protocol and making baseline assessments.

### 2.2. Study Participants

The study included post Coronary Artery Bypass Graft surgery (CABG) patients of both sex with age ranging from 45–80 years. Patients who were extubated and hemodynamically stable were also included in study if they could sit, stand, and ambulate independently. Patients were excluded from study if they had arrhythmias or any other unstable medical condition rendering them unable to perform exercise protocol of in-patient cardiac rehabilitation. The patients who were unable to participate in any exercise program because of any musculoskeletal, neurological, or psychological illness were also excluded from the study. Participants pathway during the study is reported in Figure 1.

### 2.3. Protocol

Patients were approached during in-patient phase of cardiac rehabilitation in intensive care unit. After initial anamnesis and baseline assessments for anthropometric variables, the following exercises were performed in exercise training in cardiac rehabilitation, Table 1.

Exercise tolerance was assessed through 6MWT and ISWT which were administered at 3rd, 5th, and 7th post-operative days with a gap of 30 min between both the tests to avoid fatigue. 6MWT, being easier to perform was performed before the ISWT in order to accustom the patients with exercise testing. The tests were stopped if patients experienced intolerable shortness of breath, chest pain, sweating, severe leg cramps or if the saturation dropped below 83% (24).

### 2.4. Outcome Measures

#### 2.4.1. 6 Min Walk Test

The 6-min. Walk Test (6MWT) is a sub maximal functional capacity test used for primary assessment in cardiac rehabilitation program as well as to document functional consequences after completion of the cardiac rehabilitation program [24]. Standardized protocol was followed for performance of this test where the patients were asked to walk as far as they can, on a flat surface for 6 min. However, the patients were allowed to stop or rest if needed and continue again as soon as they were able to. The distance covered by the patients was recorded in meters [25].

#### 2.4.2. Incremental Shuttle Walk Test

The Incremental Shuttle Walk Test (ISWT) is a standardized, externally paced, incremental field walking test to assess the functional ability in patients with cardiopulmonary disorders [9]. The patients were instructed to walk on beep, back and forth between two traffic cones which were placed 10 m apart. The walking speed was increased by reducing the interval between the beeps. The patients were instructed to keep pace with beep. The test was stopped when the patient cannot keep up with the pace because of dyspnea or any other symptom such as leg pain and cramping, fatigue, or chest pain. The distance covered was calculated in meters from the number of laps completed by each patient [24].

### 2.5. Global Rating of Change—As an Anchor

The Global rating of change (GRC) scales is used to calculate anchor based MCID by determining the amount of change (progress or worsening) perceived by the patient with the passage of time [26]. The GRC scales require that a person evaluate their current health status, reminiscence that status at a prior time-point, and then calculate the change between the two [26]. The magnitude of this change is then scored on a numerical or visual analogue scale. The patients were asked to rate how they find their exercise tolerance using the question: “Compared to your endurance walk test before your rehabilitation program, how would you rate your exercise tolerance now?” Answers were categorized on a 7-point Likert scale as (−3) “large deterioration”, (−2) ‘moderate deterioration’, (−1) ‘slight deterioration’, (0) ‘no change’, (1) ‘slight improvement’, (2) ‘moderate improvement’ and (3) ‘large improvement”. These categories were dichotomized into two categories of “improvement” and “No improvement” for the calculation of anchor based MCID of the patients [26].

### 2.6. Data Analysis

Descriptive statistics were calculated by computing mean and standard deviations of continuous variables and frequency and percentages of categorical variables. To estimate the MCID of 6MWT and ISWT, both distribution-based and anchor-based approaches were used. Minimal Detectable Difference (MDD), Standard Error of Measurement (SEM) and effect size methods were used to calculate distribution based MCID [27]. SEM-Based MCID was calculated using formula: SEM = σ_1_ $\surd \left(1-r\;\right)$; where σ_1_ is baseline Standard Deviation (SD) and r is the test–retest reliability coefficient [19]. Effect Size (ES) based MCID was evaluated by using the formula: (ES) = (µ_1_ − µ_2_)/σ_1_ where µ_1_ and µ_2_ are the means at baseline and follow up correspondingly, and σ1 is the SD at the baseline. MDD represents the small change in the data needs to be calculate a clinically meaningful change in function and was derived by using the formula of MDC = 1.96 × √2 × SEM [27]. Anchor based MCID was calculated using the GRC scores of patients who reported improvement or no improvement after cardiac rehabilitation. Data was analyzed on SPSS version 21. The normality of the data was assessed by using Shapiro-Wilk test and as the data was non normally distributed (*p* < 0.05) therefore nonparametric tests during interferential statistics were used.

## 3. Results

Among 92 patients who were initially enrolled in the study, 89 completed the study protocol. Two males and one female were withdrawn because of early discharge on self-request from the hospital.

### 3.1. Demographics/Patient Characteristics

Out of 89 patients, 56 were males and 33 were females. The average age of the patients was 58.5 ± 8.3 years. Mean height was 1.71 ± 0.07 m and weight were 80.6 ± 6.9 kg. Regarding BMI, 9 patients were under weight, 26 patients were overweight, the number of patients who were obese was 15 and those with normal BMI values were 33. Baseline demographic characteristics, of patients who reported “improvement” or “no improvement” in 6MWT and ISWT are shown in Table 2 and Table 3.

### 3.2. 6 Minute Walk Test

The post CABG patients performed 6MWT along with vital monitoring on 3rd, 5th^,^ and 7th day of intervention. Test parameters of 6MWT as well as the average vitals before and after test performance, level of fatigue and dyspnea after performance of 6MWT is shown in Table 4 and Table 5.

#### 3.2.1. MCID of 6MWT

##### Distribution-Based Estimation

Considering the patient’s self-assessments, SEM was 13.03 evaluated by using the formula σ_1_√ (1 − r) in which σ_1_ is baseline SD that was 75.25 and r is the test–retest reliability coefficient. Reliability of the present test-retest results (ICC) and it was 0.97 for the present study demonstrating excellent test-retest reliability. Clinically meaningful difference (MDD) for 6MWD was 36.11 and it was calculated by using the formula 1.96 × √2 × SEM.

##### Anchor-Based Estimation

After phase I (refer to Figure 1 in methods) cardiac rehabilitation, 64 participants (71.9%) rated themselves as “improved” and 25 (28.1%) as “not improved”. Average change in 6MWD in participants who classified themselves as “improved” was 71.9 (*p* < 0.01) and the mean change in 6MWD in participants who rated themselves as “not improved” was 28.1 (*p* < 0.01). To authenticate the anchors, Mann-Whitney U test designed associating change in the 6MWT between the dichotomized groups of “improved” and “not improved”. The receiver operating characteristics (ROC) curves were created by intrigue sensitivity values (true positive rate) on the y axis and 1-specificity on the *x*-axis for different changes in 6MWT for differentiating important improvement from those participants without important improvement. Considering the 6MWTD and GRC scale of day seven, the MCID of 195 m corresponded to the sensitivity of 0.98 and 1-specificity of 0.76 with 0.651 AUC (95% CI, 0.510–0.792), as reported in Table 6**.**

### 3.3. Incremental Shuttle Walk Test

The post CABG patients performed ISWT along with vital monitoring on 3rd, 5th^,^ and 7th day of intervention. Test parameters of ISWT as well as the average vitals before and after test performance, level of fatigue and dyspnea after performance of ISWT is shown in Table 7 and Table 8.

The Mann Whitney U test was used for intergroup comparison between the participants who reported “improved” and the participants who reported “not improved” and the measures of central tendency and dispersion were reported in terms of Median and inter quartile range (IQR). The findings of inter group comparison between the scores of ISWD of the respective groups showed no significance difference (*p* = 0.224) at baseline, and (*p* = 0.077) at day 05 whereas significance difference was observed (*p* < 0.05) as shown in Table 9 with median (IQR) values; 30 (15) at day 03, 40 (15) at day 05, 50 (10) at day 07 for “improved group” and 30 (14) at day 03, 30 (10) at day 05, 32 (10) at day 07 for “not improved” group, whereas the mean ranks for improved group were 43.02, 47.83, 53.37and for not improved group were 50.35, 37.33, 22.33on day 03, 05 and 07 respectively, as reported in Table 9.

#### 3.3.1. ISWT MCID

##### Distribution-Based Estimation

The values of ISWT were compared between baseline (day 03) and at the end (day 07). The data was not normally distributed (*p* < 0.05) using the Shapiro Wilk test thus *p*-value was calculated from Wilcoxin Signed Rank test. The effect size was 3.22 that was calculated by using the formula (µ_1_ − µ_2_)/σ_1_ where µ_1_ and µ_2_ are the means at baseline which was 30 and of follow up which was 45 respectively, and σ_1_ is the SD at the baseline that was 8.17 by using means of baseline and day 07 of ISWD, baseline SD for ISWD with 95% confidence interval.

Considering the patient’s self-assessments, the standard error of measurement (SEM) was 3.99 calculated by using the formula σ_1_√ (1 − r) in which σ_1_ is baseline SD that was 8.17 and r is the test—retest reliability coefficient, Reliability of the current test- retest results (ICC)and it was 0.76 for the present study. Clinically meaningful difference (MDD) for 6MWD was 9.94 and it was calculated by using the formula 1.96 × √2 × SEM.

##### Anchor-Based Estimation

After phase I (please refer to Figure 1 in the methods) cardiac rehabilitation, 65 (48.07%) rated themselves as “improved” and 24 (35.62%) as “not improved”. Mean change in ISWD in participants who classified themselves as “improved” was 48.07(*p* < 0.01) and the mean change in ISWD in participants who rated themselves as “not improved” was 35.62 (*p* < 0.01). To authenticate the anchors, Mann- Whitney U test was designed associating change in the ISWT between the dichotomized groups of “improved” and “not improved”. The receiver operating characteristics (ROC) curves were created by intrigue sensitivity values (true positive rate) on the y axis and 1- specificity on the *x*-axis for different changes in ISWT for differentiating important improvement from those participants without important improvement. Considering the ISWD and GRC scale of day seven, the MCID of 42.5 m corresponded to the sensitivity of 0.96 and 1- specificity of 0.50 with 0.849 AUC (95% CI, 0.759–0.939). Data reported in Table 10.

## 4. Discussion

The current study aims to calculate the MCID for 6MWT and ISWT after in-patient cardiac rehabilitation utilizing both distribution-based and anchor-based approaches with the intention of developing better ways of understanding and improvement in health care goals for post-CABG patients. The authors estimated anchor based MCID in 6MWD at 195m and for ISWD at 42.5m in post-CABG patients. The effect size calculated by using distribution based MCID is 1.09 (*p* < 0.001) for 6MWD and for ISWD is 3.22 (*p* < 0.001) whereas the MDD 95% and SEM are 36.11and 13.03 (ICC = 0.97) for 6MWD and for ISWD are 9.94 and 3.99 (ICC = 0.76) respectively. To researcher’s knowledge this is the first research that considered walking tests to be carried out on CABG patient to calculate the MCID of sub maximal walk tests.

### 4.1. MCID for 6MWT

The MDD estimate for 6MWT in our study is consistent with the existing literature. The MDD calculated in current study is 36.1 m similar to the MDD in Congestive Heart Failure (CHF) patients as 32.4 m [26]. However, the estimated MCID of 6MWT is 30 m in CHF patients which is far less than the MCID calculated in current study i.e., 195 m with sensitivity of 0.98 and 1- specificity of 0.76 with 0.651 AUC (95% CI, 0.510–0.792) [26]. The possible reason for this significant difference is that the post-surgical patients have a decrease in 6MWD initially because of surgery but then there is a drastic increase in distance covered during 6MWT in post operative phase after the correction of disease as reported by Chen et al. [11]. The estimated MCID of 195 m is also greater than other cardiopulmonary disease conditions. Gremeaux et al. reported the MCID of 6MWT in patients with acute coronary syndrome undergoing cardiac rehabilitation as 25 m [19]. Holland et al. reported it to be 30.5 m in patients with diffuse parenchymal lung disease [28] This is different from current study reason being the same that the patient with cardiopulmonary diseases feel meaningful improvement with a little distance as compared to post-surgical patients because the disease limits the walking distance. This is supported by Yuksel et al. who justified the use of different values of MCID for surgical and non-surgical patients [29]. The 6MWD of the post CABG patients improved from 294.2 ± 75.2 m in post operative period to 376.5 ± 92.5 m. This was consistent with the finding of Chen et al. where the distance improved from 277.3 ± 85.7 m to 378.1 ± 95.2 m post operatively in cardiac surgery patients [11].

### 4.2. MCID for ISWT

According to our knowledge there are very few studies that have calculated MCID of ISWT in cardiac patients. In the present study, the estimated MCID of ISWT using ROC (anchor based MCID) after in patient CR was 42.5 m which is less than the MCID of ISWT calculated following cardiac rehabilitation as 70 m [9]. Houchen-Wolloff et al., reported that estimated the distribution based MCID of ISWT in participants of cardiac rehabilitation to be 36.65 m [9]. The calculated effect size was small (0.38) for overall change [9]. However, the effect size yielded in current study is 3.22 (*p* < 0.001). There was poor agreement between distribution and anchor based MCID as also reported in previous studies [3,9]. The calculated MCID was 47.5 in COPD patients which is very close to 42.5 m as calculated by the current study [30]. There were 65 (48.07%) patients who rate themselves as improved and the distance travelled from baseline till day 07 justify their perception. The mean distance improved about 27 m in seven days in the current study as compared to the 65.2 m in six weeks in a study conducted by Houchen-Wolloff et al. on patients after cardiac rehabilitation [9]. The patients who rated themselves as “not improved” were 24 (35.62%), though among them few patients were those that have improved up to few meters. This shows that patients were unable to rate small changes in their performance. This might be due to the Hawthorne effect where the patients rate differently when they know that they are being observed or because of high expectations for improvement in their exercise tolerance after surgery which is not observed on 7th post operative day [9].

## 5. Limitations

The values are recorded solemnly on the patient’s performance. Certain factors including parametric parameters like obesity, type of drugs used, and amount of supplemental oxygen were not considered. The pre-operative physical activity of the patient was not assessed which can affect the post-operative physical performance According to researcher’s knowledge, there is no study that has calculated MCID in outpatient phases of Cardiac rehabilitation so future studies should be made to estimate clinically meaningful change in outpatient phases of cardiac rehabilitation.

## 6. Conclusions

The current study aims to calculate the MCID for 6MWT and ISWT after in-patient cardiac rehabilitation utilizing both distribution-based and anchor-based approaches with the intention of developing better way of understanding and improvement in health care goals for post-CABG patients. The results provide initial estimates of MCID of 6MWT and ISWT in post cardiac surgery patients. The study supports the use of these objective measurements (i.e., 6MWT and ISWT) tests in post CABG patient and will help clinicians and researchers to interpret changes and set goals for cardiac rehabilitation.

## Figures and Tables

**Figure 1 ijerph-19-14270-f001:**
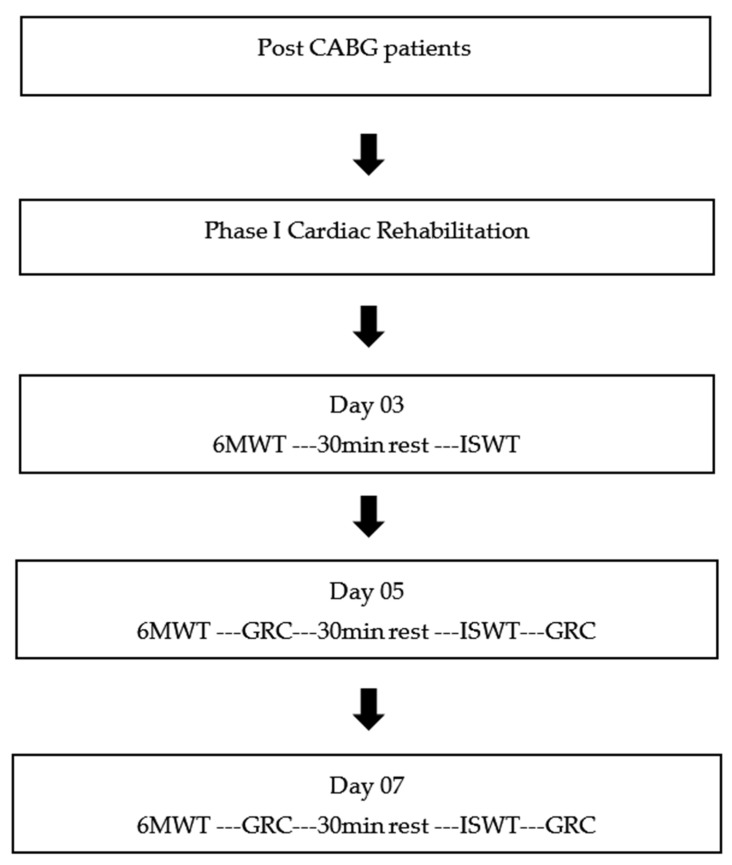
Flow diagram for the patient pathway through study. CABG = Coronary Artery Bypass graft surgery 6MWT = 6 min walking test; GRC = Global Rating of Change; ISWT = Incremental Shuttle Walk Test.

**Table 1 ijerph-19-14270-t001:** Table showing the exercise plan for patients during in-patient cardiac rehabilitation.

Exercise Program	Specifications
Deep breathing exercises	10 repetitions twice a day
Nebulization, chest percussions and huffing & coughing	Twice a day
Active range of motion exercises with minimal resistance	10 reptations, twice a day
Incentive spirometry	15 repetitions after every 2 h
Walk	with/without walking aids
Climbing one flight of the stairs (4 Steps)	on 4th post operative day

List of exercises performed in standard cardiac rehabilitation.

**Table 2 ijerph-19-14270-t002:** Frequencies and percentages of demographic characteristics of patients reporting “improvement” and “no improvement” in 6MWT and ISWT.

Variable	Category	Patients (*n* = 89) A	6MWT		ISWT	
			Patients Reporting “No Improvement” (*n* = 64)	Patients Reporting “Improvement” (*n* = 25)	Patients Reporting “No Improvement” (*n* = 6)	Patients Reporting “Improvement” (*n* =8 3)
Sex	Males	62.9%	68%	60.9%	70.8%	60.0%
	Females	37.1%	32%	39.1%	29.2%	40.0%
Diagnosis	TVCAD	24.7%	24.0%	25%	37.5%	20.0%
	Stenosis of LAD	52.8%	52.0%	53.1%	45.8%	55.4%
	Stenosis of proximal Cx artery	15.7%	16.0%	15.6%	12.5%	16.9%
	Others	6.7%	8.0%	6.3%	4.2%	7.7%
Diabetics	Positive	47.2%	48%	46.9%	58.3%	43.1%
	Negative	52.8%	52%	53.1%	41.7%	56.9%
HTN	Positive	59.6%	64%	57.8%	58.3%	60.0%
	Negative	40.4%	36%	42.2%	41.7%	40.0%
BMI	Under weight	10.1%	8%	10.9%	16.7%	7.7%
	Normal	41.6%	32%	45.3%	37.5%	43.1%
	Overweight	29.2%	32%	28.1%	25.0%	30.8%
	Obese	19.1%	28%	15.6%	20.8%	18.5%
Use of supplemental oxygen	No	86.5%	100%	81.3%	91.7%	84.6%
Type of Graft	Venous	41.6%	36%	43.8%	45.8%	40.0%
	Arterial	58.4%	64%	56.3%	54.2%	60.0%
Type of CABG	Complete CABG	60.7%	48%	65.6%	75.0%	55.4%
	MIDCABG	39.3%	52%	34.4%	25.0%	44.6%

TVCAD = triple vessel coronary artery disease, LAD = Left anterior descending, Cx = circumflex artery, MIDCAB = Minimally invasive direct coronary artery bypass; HTN = Hypertension; BMI = Body mass index; CABG = Coronary artery bypass graft surgery;6MWT = 6 min walking test; ISWT = Incremental Shuttle Walk Test.

**Table 3 ijerph-19-14270-t003:** Mean and standard deviation of anthropometric measures and surgical history of participants reporting “improvement” and “no improvement” in 6MWT and ISWT.

Variables	Patients (*n* = 89)	6MWT		ISWT	
		Patients Reporting “No Improvement” (*n* = 64)	Patients Reporting “Improvement” (*n* = 25)	Patients Reporting “No Improvement” (*n* = 6)	Patients Reporting “Improvement” (*n* = 83)
Age (years)	58.4 ± 8.3	57.40 ± 7.78	58.89 ± 8.5	57.70 ± 8.06	58.75 ± 8.45
Height (meters)	1.7 ± 0.07	1.71 ± 0.07	1.70 ± 0.06	1.72 ± 1.70	67.18 ± 0.07
Weight (kg)	80.6 ± 6.9	79.56 ± 7.2	81.07 ± 6.7	80.54 ± 7.05	80.69 ± 6.93
Ejection Fraction (%)	58.8 ± 5.9	60.36 ± 6.1	58.31 ± 5.7	59.45 ± 5.83	58.67 ± 5.98
Duration of Surgery (hours)	3.7 ± 0.52	3.72 ± 0.54	3.78 ± 0.51	3.87 ± 0.44	3.72 ± 0.54
Duration of ICU stay in (days)	2.0 ± 0.0	2.0 ± 0.0	2.0 ± 0.0	2.00 ± 0.0	2.00 ± 0.0
Duration of arousal from anesthesia (hours)	3.1 ± 0.74	3.28 ± 0.79	3.14 ± 0.73	3.00 ± 0.72	3.24 ± 0.75

ICU = Intensive care unit, 6MWT = 6 minutes walking test; ISWT = Incremental Shuttle Walk Test.

**Table 4 ijerph-19-14270-t004:** Descriptive data for vitals, fatigue, and dyspnea of patients during 6MWT on day 3, 5 and 7 of cardiac rehabilitation.

		Day 03		Day 05		Day 07	
Vitals after 6MWT							
		Pre-Test (Mean ± SD)	Post Test (Mean ± SD)	Pre-Test (Mean ± SD)	Post Test (Mean ± SD)	Pre-Test (Mean ± SD)	Post Test (Mean ± SD)
Heart rate (b/m)		73.78 ± 6.5	84.74 ± 5.75	73.48 ± 6.02	82.76 ± 5.20	72.89 ± 5.70	79.57 ± 2.63
Respiratory rate (b/m)		16.70 ± 0.89	25.95 ± 2.52	16.91 ± 0.82	25.42 ± 2.76	16.73 ± 0.93	25.22 ± 1.72
Blood Pressure	Systolic (mmHg)	133.21 ± 8.9	164 ± 18.18	133.48 ± 9.30	150.58 ± 41.11	133.48 ± 9.30	155.50 ± 24.02
	Diastolic (mmHg)	86.62 ± 2.4	94.17 ± 2.88	86.78 ± 2.22	90.49 ± 4.62	86.55 ± 2.21	90.08 ± 4.43
SPO_2_(%)		92.16 ± 4.0	80.66 ± 6.13	92.22 ± 4.43	87.84 ± 4.54	93.73 ± 3.63	91.74 ± 4.72
**Fatigue after 6MWT**							
No fatigue at all		0 (0%)		0 (0%)		0 (0%)	
Slightly Fatigue		17 (19.1%)		50 (56.2%)		68 (76.4%)	
Moderately fatigue		56 (62.9%)		32 (36.0%)		14 (15.7%)	
Severe Fatigue		16 (18.0%)		7 (7.9%)		7 (7.9%)	
**Dyspnea after 6MWT**							
No Breathlessness At all		0 (0%)		0 (0%)		0 (0%)	
Very very slight breathlessness		0 (0%)		0 (0%)		1 (1.1%)	
Very Slight breathlessness		0 (0%)		0 (0%)		19 (21.3%)	
Slight Breathlessness		0 (0%)		1 (1.1%)		1 (1.1%)	
Moderate Breathlessness		38 (42.7%)		36 (40.4%)		35 (39.3%)	
Somewhat Severe Breathlessness		36 (40.4%)		39 (43.8%)		15 (16.9%)	
Severe Breathlessness		9 (10.1%)		12 (13.5%)		6 (6.7%)	
Maximum Breathlessness		6 (6.7%)		1 (1.1%)		12 (13.5%)	

SPO2 = Oxygen saturation; 6MWT = 6 min walking test; ISWT = Incremental Shuttle Walk Test.

**Table 5 ijerph-19-14270-t005:** Descriptive data for test parameters of 6MWT.

Variables		Day 03 (Mean ± SD)	Day 05 (Mean ± SD)	Day 07 (Mean ± SD)
No. of Laps		4.90 ± 1.25	5.60 ± 1.37	6.25 ± 1.53
Total Distance covered (meters)		294.26 ± 75.25	336.50 ± 82.59	376.51 ± 92.57
Time Taken (minutes)		5.68 ± 0.76	5.77 ± 0.82	5.91 ± 0.45
No. of Times the patients stopped during test		1.60 ± 0.54	1.33 ± 0.50	1.33 ± 0.51
GRC after 7 days of intervention	Improvement			64 (71.9%)
	No improvement			25 (28.1%)

SD = Standard Deviation; GRC = Global rate of change.

**Table 6 ijerph-19-14270-t006:** Showing Distribution and Anchor based estimation of Minimal Clinically Important Difference in 6MWD in post-CABG patients.

Distribution Based Estimation (*n* = 89)					
6mwd at Baseline	Baseline SD	ICC		SEM	MDD
294.26	75.25	0.97		13.03	36.11
**Anchor Based Estimation (*n* = 89)**					
Estimate of MCID for change in 6MWD (meters)			195 m		
AUC (95% CI)			0.651 (0.510–0.792)		
Sensitivity			0.98		
1-Specificity			0.76		

**Table 7 ijerph-19-14270-t007:** Descriptive data for vitals, fatigue, and dyspnea of patients during ISWT on day 3, 5 & 7.

		Day 03		Day 05		Day 07	
Vitals after 6MWT							
		Pre-Test (Mean ± SD)	Post Test (Mean ± SD)	Pre-Test (Mean ± SD)	Post Test (Mean ± SD)	Pre-Test (Mean ± SD)	Post Test (Mean ± SD)
Heart rate (b/m)		73.31 ± 6.4	84.66 ± 5.7	73.33 ± 6.03	83.24 ± 5.4	73.12 ± 5.6	81.82 ± 6.7
Respiratory rate (b/m)		16.60 ± 0.85	26.53 ± 2.43	17.04 ± 0.79	24.87 ± 2.5	16.66 ± 0.95	25.32 ± 1.3
B.P	Systolic (mmHg)	132.65 ± 10.3	174.26 ± 14.05	133.48 ± 9.3	168.32 ± 19.8	134.04 ± 9.7	152.15 ± 21.6
	Diastolic (mmHg)	86.62 ± 2.4	94 ± 3.09	86.86 ± 2.2	91.44 ± 4.5	87.10 ± 1. 9	90.08 ± 4.43
SO_2_(%)		92.16 ± 4.09	89.56 ± 5.23	92.48 ± 4.02	88.78 ± 4.7	92.90 ± 3.4	88.57 ± 4.2
**Fatigue after 6MWT**							
No fatigue at all		0 (0%)		0 (0%)		0 (0%)	
Slightly Fatigue		20 (22.5%)		27 (30.3%)		55 (61.8%)	
Moderately fatigue		34 (38.2%)		57 (64%)		34 (38.2%)	
Severe Fatigue		35 (39.3%)		5 (5.6%)		0 (0%)	
**Dyspnea after 6MWT**							
No Breathlessness At all		0 (0%)		0 (0%)		0 (0%)	
Very Very slight breathlessness		0 (0%)		0 (0%)		3 (3.4%)	
Very Slight breathlessness		0 (0%)		0 (0%)		25 (28.1%)	
Slight Breathlessness		0 (0%)		1 (1.1%)		4 (4.5%)	
Moderate Breathlessness		38 (42.7%)		36 (40.4%)		27 (30.3%)	
Somewhat Severe Breathlessness		39 (43.8%)		39 (43.8%)		6 (6.7%)	
Severe Breathlessness		10 (11.2%)		12 (13.5%)		1 (1.1%)	
Maximum Breathlessness		2 (2.2%)		1 (1.1%)		23 (25.8%)	

SPO_2_ = Oxygen saturation; 6MWT = 6 min walking test; ISWT = Incremental Shuttle Walk Test.

**Table 8 ijerph-19-14270-t008:** Showing descriptive data of test parameters of ISWT.

Variables		Day 03 (Mean ± SD)	Day 05 (Mean ± SD)	Day 07 (Mean ± SD)
No. of Laps		3.11 ± 0.81	3.80 ± 0.82	4.47 ± 0.97
Total Distance covered (meters)		31.12 ± 8.17	38.08 ± 8.24	44.71 ± 9.78
Time Taken (seconds)		54.26 ± 10.54	62.47 ± 7.72	81.46 ± 8.05
No. of Times the patients stopped during test		1.50 ± 0.57	1.0 ± 0.0	2.0 ± 0.0
GRC scale after 7 days of intervention	Improvement			65 (48.07%)
	No improvement			24 (35.62%)

SD = Standard Deviation; GRC = Global rate of change.

**Table 9 ijerph-19-14270-t009:** Between groups comparison using Mann- Whitney U test showing median, mean rank, Z value and P value of distance covered at day 03, 05 and 07 of ISWT.

	Median (IQR)		Mean Rank		Z Value	*p* Value
	Improvement (*n* = 65)	No Improvement (*n* = 24)	Improvement (*n* = 65)	No Improvement (*n* = 24)		
ISWD at day 03	30 (15)	30 (14)	43.02	50.35	−1.21	0.224
ISWD at day 05	40 (15)	30 (10)	47.83	37.33	−1.76	0.077
ISWD at day 07	50 (10)	32 (10)	53.37	22.33	−5.16	0.000

ISWD = Incremental shuttle walk distance, IQR = Interquartile range.

**Table 10 ijerph-19-14270-t010:** Showing Distribution and Anchor based estimation of Minimal Clinically Important Difference in ISWD in post-CABG patients.

Distribution Based Estimation (*n* = 89)					
ISWD at Baseline	Baseline SD	ICC		SEM	MDD
31.12	8.17	0.76		3.99	9.94
**Anchor based estimation (*n* = 89)**					
Estimate of MCID for change in ISWD (meters)			42.5 m		
AUC (95% CI)			0.849 (0.759–0.939)		
Sensitivity			0.70		
1-Specificity			0.20		

6MWD = 6-min Walk distance; SD = Standard deviation; ICC = Intraclass correlation coefficient; SEM = Standard error of measurement; MDD = Minimal detectable difference; MCID = Minimal clinically important distance; AUC = Area under curve.

## Data Availability

Data are available contacting the corresponding author.

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
