# Peer review of "Clinically Meaningful Change in 6 Minute Walking Test and the Incremental Shuttle Walking Test following Coronary Artery Bypass Graft Surgery"

_ijerph, 2022, doi:10.3390/ijerph192114270_

Round 1

Reviewer 1 Report

On the whole this is a very well written study. I just have some minor amendments to suggest. But do go through and check the grammar are there several minor mistakes throughout which could be improved. 

Abstract

Incremental walk test needs shuttle included to make sense of the abbreviation.

I would write in the past tense with regards to the statistics (is should be was).

Intro

P2L50 – delete a in front of submaximal

P2L69 – correct Busch et al citation

PL77 – and should be the?

Did you randomise the 6MWT with the ISWT or was the 6MWT always done first? If so, please justify.

You said 83 patients were sufficient for power but you recruited 92 – why? I assume to account for drop out, but this would be good to be noted when talking about power. You aimed to recruit X number to account for drop out which is estimated to be 20% for this type of study?

How was distance measured in the 6 min walk test? Accelerometer?

Check et al when citing – it keeps going to et all

There should be a full stop after al.

Author Response

The authors would like to thank all the reviewers for taking the time to produce high-quality comments and considerations that improved the quality and validity of the manuscript. We have produced a table with your comments, our replies and where to find corrections and modifications in the texts. The manuscript has also been modified using Track Changes. We hope this can help you navigate through the document.

Following your comments and corrections, we believe the manuscript has improved its quality and validity. Therefore we favour the publication of “Clinically meaningful changes in 6 Minute Walking Test and the Incremental Shuttle Walking Test following Coronary Artery Bypass Graft surgery” on MDPI as we believe this can be an important step for setting goals related to functional capacity in patients after coronary artery bypass graft surgery.

We are looking forward to your further comments and considerations.

Thank you for your time.

Best wishes

Suman Sheraz

Reviewer 1

N

Comments

Reply

Page / Lines

Abstract:

1

Incremental walk test needs shuttle included to make sense of the abbreviation.

Updated

Pg 1 Line 11

2

I would write in the past tense with regards to the statistics (is should be was).

Updated

Pg 1 line 23-25

Introduction

3

P2L50 – delete a in front of submaximal

Updated

Pg 2 Line 50

4

P2L69 – correct Busch et al citation

Updated

Pg 2 Line 69

5

PL77 – and should be the?

The authors are not sure about what this comment mean. Can you please provide more details?

Pg 2 Line 77

5

Did you randomise the 6MWT with the ISWT or was the 6MWT always done first? If so, please justify.

Being sub maximal test, 6MWT was done first to accustomize the patient with exercise testing. Additional details have been included in the text

Pg 4, Line 125, 126

6

You said 83 patients were sufficient for power but you recruited 92 – why? I assume to account for drop out, but this would be good to be noted when talking about power. You aimed to recruit X number to account for drop out which is estimated to be 20% for this type of study?

Yes, to account for dropouts as much as available patients were recruited.

Pg 2, Lines 85-87

7

How was distance measured in the 6 min walk test? Accelerometer?

No, 30m distance was marked via two cones at each end. Every 3m was marked with a colored tape on the floor.

8

Check et al when citing – it keeps going to et all, there should be a full stop after al.

Updated

Reviewer 2 Report

The authors present an useful study about assessment of the early results of cardiac rehabilitation after CABG surgery. I have only minor comments: a) it would be more illustrative to add statistical significance of difference even in tables with descriptive values 4,5,7 and 8

b) as authors stated in limitations paragraph, it will be very useful to assess results from outpatient cardiac rehabilitation (CR), and/or long term outcomes from follow-up of patients undergoind in- and outpatient CR, and the relation of these outcomes to values obtained e.g. from ISWT and 6MWT.

Author Response

The authors would like to thank all the reviewers for taking the time to produce high-quality comments and considerations that improved the quality and validity of the manuscript. We have produced a table with your comments, our replies and where to find corrections and modifications in the texts. The manuscript has also been modified using Track Changes. We hope this can help you navigate through the document.

Following your comments and corrections, we believe the manuscript has improved its quality and validity. Therefore we favour the publication of “Clinically meaningful changes in 6 Minute Walking Test and the Incremental Shuttle Walking Test following Coronary Artery Bypass Graft surgery” on MDPI as we believe this can be an important step for setting goals related to functional capacity in patients after coronary artery bypass graft surgery.

We are looking forward to your further comments and considerations.

Thank you for your time.

Best wishes

Suman Sheraz

Reviewer 2

N

Comments

Reply

Page / Lines

It would be more illustrative to add statistical significance of difference even in tables with descriptive values 4,5,7 and 8

We appreciate your comments and will consider this for future studies. However, for the current document we reported descriptive data provided for test parameters and vitals during 6MWT / ISWT as we needed to consider the response of patients during exercise testing via these tests.

As authors stated in limitations paragraph, it will be very useful to assess results from outpatient cardiac rehabilitation (CR), and/or long term outcomes from follow-up of patients undergoind in- and outpatient CR, and the relation of these outcomes to values obtained e.g. from ISWT and 6MWT.

The authors agreed with the reviewer comment and have included outpatients consideration in the limitations

Pg 14, Line